# Lipidomic Characterization of Whey Concentrates Rich in Milk Fat Globule Membranes and Extracellular Vesicles

**DOI:** 10.3390/biom14010055

**Published:** 2023-12-31

**Authors:** Richard R. Sprenger, Marie S. Ostenfeld, Ann Bjørnshave, Jan T. Rasmussen, Christer S. Ejsing

**Affiliations:** 1Department of Biochemistry and Molecular Biology, VILLUM Center for Bioanalytical Sciences, University of Southern Denmark, 5230 Odense, Denmark; 2Arla Foods Ingredients Group P/S, 8260 Viby J, Denmark; 3Department of Molecular Biology and Genetics, Aarhus University, 8000 Aarhus, Denmark; 4Cell Biology and Biophysics Unit, European Molecular Biology Laboratory, 69117 Heidelberg, Germany

**Keywords:** lipidomics, milk fat globule membranes, extracellular vesicles, whey fat concentrate, whey protein concentrate

## Abstract

Lipids from milk fat globule membranes (MFGMs) and extracellular vesicles (EVs) are considered beneficial for cognitive development and human health. Milk-derived whey concentrates rich in these lipids are therefore used as ingredients in infant formulas to mimic human milk and in medical nutrition products to improve the metabolic fitness of adults and elderly people. In spite of this, there is no consensus resource detailing the multitude of lipid molecules in whey concentrates. To bridge this knowledge gap, we report a comprehensive and quantitative lipidomic resource of different whey concentrates. In-depth lipidomic analysis of acid, sweet, and buttermilk whey concentrates identified 5714 lipid molecules belonging to 23 lipid classes. The data show that the buttermilk whey concentrate has the highest level of fat globule-derived triacylglycerols and that the acid and sweet whey concentrates have the highest proportions of MFGM- and EV-derived membrane lipids. Interestingly, the acid whey concentrate has a higher level of cholesterol whereas sweet whey concentrate has higher levels of lactosylceramides. Altogether, we report a detailed lipid molecular compendium of whey concentrates and lay the groundwork for using in-depth lipidomic technology to profile the nutritional value of milk products and functional foods containing dairy-based concentrates.

## 1. Introduction

Milk and milk products are important sources of dietary lipids [1,2,3]. This because milk lipids can serve as nutrients for energy metabolism and as important structural building blocks for tissue growth during infancy and childhood as well as for tissue maintenance and repair during adulthood. Milk-derived whey concentrates, that typically result as side-products of industry-scale cheese production, also contain substantial amounts of milk lipids. This type of milk product can be used for production of functional foods as it allows harnessing the beneficial effects and high nutritional value of milk lipids. As such, whey concentrates are used as ingredients in a variety of dietary and nutritional products, including infant formulas to mimic the properties of human milk as well as medical nutrition products that improve the metabolic fitness and cognitive function of adults and elderly people [4,5,6]. Despite the great interest in the nutritional value of milk lipids, and the use of these to functionalize and valorize foods, there is general discrepancy in the knowledge about the molecular structure and composition of lipids in milk products and derived products. This is because routine analyses of milk products typically rely on bulk measurements of total amounts of neutral lipids (i.e., fat) and phospholipids. This is in sharp contrast to the molecular depth that can be achieved using state-of-the-art lipidomic technology where individual lipid molecules with distinct hydrocarbon chains can be identified and accurately quantified [7].

Lipid molecules in milk are primarily present as two types of colloidal particles, namely milk fat globules (MFGs) and extracellular vesicles (EVs) [8,9]. MFGs are the most abundant and composed of a triacyclglycerol (TAG)-rich core surrounded by a membrane monolayer and membrane bilayer (i.e., together a membrane trilayer), which collectively are termed the milk fat globule membrane (MFGM). MFGs are generally thought to be secreted by a mechanism where intracellular lipid droplets, together with their monolayer membrane (derived from the endoplasmic reticulum (ER) and rich in glycerophospholipids), become enveloped by the apical plasma membrane (rich in sphingolipids and cholesterol) and shredded off from milk-producing cells [10]. In comparison, EVs are composed of a single membrane bilayer, which originates from shredding off the apical plasma membrane (rich in sphingolipids and cholesterol, and devoid of TAGs) as microvesicles or secretion of intracellularly formed exosomes [11,12]. Notably, cream separation of milk yields a cream-phase rich in MFGs as well as associated MFGMs, and a skimmed milk fraction containing EVs and MFG-derived MFGMs. Here, it is estimated that the 40–80% of all phospholipids in milk end up in the skimmed milk, whereof 20–40% and 60–80% derive from EVs and MFGMs, respectively [13,14].

The biosynthesis of individual lipid molecules adds another level of complexity to the biogenesis and nutritional value of milk lipids. The overall lipid constituents of milk are known [15,16,17] and can be readily explained by the lipid biosynthetic machinery of milk-producing cells. For example, TAG molecules are produced via a series of enzyme-catalyzed reactions using glucose and exogenous fatty acyl (FA) chains as precursors. More specifically, glucose is converted to glycerol-phosphate, which is then combined with two FA chains taken up from the circulation or produced endogenously using glucose or other metabolic precursors [18,19]. The produced intermediate, termed phosphatidic acid, can be dephosphorylated and combined with a third FA chain to yield a TAG molecule. Alternatively, phosphatidic acid can be used for synthetizing other glycerophospholipids, including phosphatidylcholines (PC), phosphatidylethanolamines (PEs), and phosphatidylinositols (PIs), which produce the membrane monolayer and bilayers of MFGs and EVs. Moreover, milk-producing cells also produce sphingolipids, including sphingomyelin (SM) and glycosphingolipids [15]. Notably, milk-producing cells can produce a wide array of individual lipid molecules by combining a multitude of available FA chains that are taken up from the circulation or produced endogenously. For example, it is known that TAG molecules in cow milk is made up of a combination of even- and odd-numbered saturated and monounsaturated FA chains, including 4:0, 15:0, 16:0, and 17:1 [20]. A similar combinatorial complexity exists for hydrocarbon chains of glycerophospholipids and sphingolipids [21].

Here, we set out to generate a comprehensive, quantitative resource of individual lipid molecules in cow milk-derived whey concentrates. To this end, we performed in-depth MS^ALL^ lipidomic analysis and compared the molecular compositions of three different types of whey concentrates having different proportions of MFGs, MFGMs, and EVs. Altogether, the present report disseminates a detailed molecular compendium featuring molar abundances of several thousand lipid molecules. Furthermore, our work lays the groundwork for using in-depth lipidomic technology for profiling the nutritional value of milk products and functional foods containing dairy-based concentrates.

## 2. Materials and Methods

### 2.1. Chemicals

Methanol, 2-propanol and water were purchased from Biosolve BV (Valkenswaard, The Netherlands). Chloroform was from Rathburn Chemicals (Walkerburn, UK). Ammonium formate was from Sigma-Aldrich (Buchs, Switzerland). All solvents and chemicals were of the highest analytical grade. Synthetic lipid standards were purchased from Avanti Polar Lipids (Alabaster, AL, USA).

### 2.2. Procurement of Whey Concentrates

Lacprodan MFGM-10 (termed sweet whey concentrate) was provided by Arla Food Ingredients (Viby J, Denmark). The concentrate was produced in October 2020 using milk from Danish Holstein cows. The acid whey concentrate was obtained following acidic precipitation of proteins in a micellar casein fraction derived from skimmed milk, removal of precipitated proteins, concentration of the remaining soluble fraction, and spray-drying of the resulting whey. The concentrate was produced in November 2020 using milk from Danish Holstein cows. The buttermilk whey concentrate was obtained from a side stream of butter production. Sweet buttermilk, the liquid left behind after butter is churned from cream, was subjected to acidic precipitation of proteins, removal of precipitated proteins, and spray-drying of the resulting whey. The concentrate was produced in January 2021 using milk from Danish Holstein cows. The concentrates were analyzed for total amount of phospholipids (i.e., glycerophospholipids and sphingomyelin) by ^31^P-NMR spectroscopy by Spectral Service AG (Köln, Germany), as well as the total amounts of lipid and protein by Eurofins Steins Laboratorium A/S (Vejen, Denmark).

### 2.3. Mass Spectrometry-Based Lipidomics

The whey concentrates, approximately 20 mg, were dissolved in 1 mL H_2_O. These suspensions were further diluted in H_2_O to yield a final concentration of 1 mg/mL. Aliquots corresponding to 60 μg acid whey concentrate, 35 μg sweet whey concentrate, and 20 μg buttermilk whey concentrate were added to 155 mM ammonium formate buffer to yield a total volume of 200 μL and spiked with 30 μL internal lipid standard mixture dissolved in chloroform/methanol (1:10, *v*/*v*). Next, lipid extraction was carried out by adding 990 μL cold chloroform/methanol (2:1, *v*/*v*), vigorous shaking (1400 rpm, 4 °C) for 1.5 h, followed by centrifugation (1000× *g*, 4 °C) for 2 min to promote phase separation. Lower organic phases containing extracted lipids were collected in new vials. These extracts were dried using a SpeedVac and dissolved in 100 μL chloroform/methanol (1:2, *v*/*v*). Lipid extracts (10 μL) were loaded in a 96-well plate and added 12.9 μL 13.3 mM ammonium formate in 2-propanol or 12.9 μL 1.33 mM ammonium formate in 2-propanol for mass spectrometric analysis in positive or negative ion mode, respectively, followed by covering the 96-well plate with sealing tape to avoid evaporation of the organic solvents.

The lipid extracts were analyzed by MS^ALL^ analysis in positive and negative ion mode using an Orbitrap Fusion Tribrid (Thermo Fisher Scientific, Waltham, MA, USA) equipped with a robotic nanoflow ion source, TriVersa NanoMate (Advion Biosciences, Ithaca, NY, USA). High-resolution survey Fourier transform MS (FTMS^1^) spectra were recorded across the range of *m*/*z* 280 to 1400 using a max injection time of 100 ms, automated gain control at 1 × 10^5^, three microscans and a target resolution of 500,000. Consecutive FTMS^2^ spectra were acquired for all precursors in the range of *m*/*z* 398.3 to 1000.8 in steps of 1.0008 Da, recorded across a range starting from *m*/*z* 150 until the precursor *m*/*z* value +10 Da, using max injection time of 100 ms, automated gain control at 5 × 10^4^, one microscan, a target resolution of 30,000, HCD fragmentation and a quadrupole ion isolation width of 1.0 Da [7]. Targeted monitoring of ammoniated cholesterol (*m*/*z* 404.38869) and its internal standard cholesterol + ^2^H_7_ (*m*/*z* 411.43265) was done by MSX analysis using max injection time of 600 ms, automated gain control at 5 × 10^4^, five microscans, a target resolution of 120,000, HCD fragmentation at 8%, and a quadrupole ion isolation width of 1.5 Da for each precursor [22].

### 2.4. Lipid Quantification from Mass Spectrometric Data

Identification and quantification of lipid molecules was done using ALEX^123^ software and a data processing pipeline in SAS 9.4 (SAS Institute, Cary, NC, USA) [23,24,25]. Details specifying which polarities, adduct ions and internal standards were used for identification and quantification of different lipid classes are summarized in Appendix A. Briefly, lipid molecules detected by full-scan FTMS^1^ were identified using a maximum *m*/*z* tolerance of ±0.0040 amu, corrected for potential ^13^C isotope interference, required to have a relative detection frequency greater than 0.66 (equivalent to being detected in 66% of all biological replicates for a given sample group), and reported at the “species-level”. Lipid fragment ions detected by FTMS^2^ were identified using a maximum *m*/*z* tolerance of ±0.0065 amu, required to have a relative detection frequency greater than 0.4 (equivalent to being detected in 40% of all replicates for a given sample group), and reported as “molecular lipid species-specific fragments” (MLF) or “lipid class-specific fragments” (LCF) [24,25]. For identification of lipids reported at the species-level (e.g., SM 34:1;2), at least one confirmatory LCF detected by FTMS^2^ was required. For identification of molecular lipid species identified by detection of MLFs, the following criteria were set: (i) asymmetric molecular lipid species must be detected by at least a complementary pairs of MLFs (except for protonated PE O- species that do not release abundant complementary MLFs); (ii) the molecular lipid species must have an ALEX score >0.5 (calculated as the number of detected MLFs relative to the total number of MLFs available in the ALEX^123^ database) or an ALEX score ≤ 0.5 but with detection of >2 MLFs (with the exception that protonated PE O- species could be detected by at least 2 MLFs); and (iii) confirmation by detection of the corresponding lipid molecule at the species-level by full-scan FTMS^1^ analysis. Identified lipid molecules were quantified (i.e., pmol) by normalizing their intensities to that of respective internal lipid standards, subsequent multiplication by the amount of the respective lipid standard and normalization to the extracted sample amount (i.e., µg concentrate). Furthermore, the molar values of individual lipid molecules were also normalized to the total of all lipids to yield the relative unit mol% per all monitored lipids. Other relative units were computed similarly.

### 2.5. Statistical Analysis

Statistical analyses, specifically analysis of variance (ANOVA) with multiple hypothesis correction, were carried out using SAS 9.4. Visual inspection of data quality and lipidomic data were done using Tableau Desktop version 2020.2.7 (Tableau Software). Displayed data represent mean ± SD (*n* = 6 replicates for acid whey concentrates and *n* = 3 for the other concentrates).

### 2.6. Lipid Nomenclature

Lipid classes are denoted by their class abbreviations [24,26,27]. For lipids reported at the “species-level”, the combined number of carbons, double bonds, and hydroxyl-groups in the hydrocarbon chains is indicated after the lipid class abbreviation. For example, “PE 38:4” denotes a PE lipid with thirty-eight carbons and four double bonds spread across both individual FA chains. For lipids reported at the “molecular species-level” (i.e., identification of individual FA chain compositions), individual hydrocarbon chains are indicated in the format of ‘total number of carbons:number of double bonds’, with individual FAs separated by a dash for glycerolipids and glycerolipids. For example, “SM 18:1;2/23:0” indicates an SM lipid containing a C_18_-sphingosine chain having one double bond and two hydroxyl-groups (i.e.,18:1;2) and an amide-linked 23:0 acyl chain. For ether lipids, ether-bound hydrocarbon chains are preceded with an “O” indicating either 1-O-alkyl ether or 1-O-alkenyl ether (plasmalogen) linkage. For example, PE O-18:1/20:4 is a PE O- lipid with a 20:4 FA chain and an 18-carbon ether-linked chain with one double bond. The double bond can be either that of a 1-O-alkenyl ether or positioned along the remainder of the FA chain as a 1-O-alkyl ether. It is noted that exact *sn-* positions and locations of double bonds of individual FAs is not resolved by the applied lipidomic technology.

## 3. Results and Discussion

### 3.1. Bulk Composition of Whey Concentrates

In this study, we profiled the molecular lipid compositions of three milk-based whey protein–lipid concentrates. The first concentrate was an acid whey concentrate obtained from a micellar casein fraction. The second was a sweet whey concentrate obtained from cheese production. The third was a buttermilk whey concentrate obtained from a side stream of butter production. Their relative amounts of total lipid, phospholipid, fat (assumed to be mainly TAG), protein, and dry matter, estimated using conventional gravimetric analysis and NMR spectroscopy, are shown in Table 1. Overall, the buttermilk whey concentrate has the highest proportion of fat (25%), which is followed by the sweet whey (14%) and acid whey (6%). The sweet whey, acid whey and buttermilk whey have comparable proportions of phospholipids (8%, 5%, and 9%, respectively). Using the relative proportions of total fat and phospholipid as proxies for lipid origins suggest that the buttermilk whey concentrate is richer in TAG-rich cores derived from MFGs (74% total fat) and has a lower proportion of membranes from MFGMs and EVs (26% phospholipid). In comparison, lipids in the acid whey concentrate stem from membranes of MFGMs and EVs (46% phospholipid) and do so to a higher extent than the sweet whey concentrate (35% phospholipid). Conversely, the sweet whey concentrate has a higher proportion from TAG-rich cores (65% total fat) as compared to the acid whey concentrate (54% total fat).

### 3.2. Comprehensive In-Depth Lipid Profiling of Whey Concentrates

Samples of each concentrate, prepared by dissolving these in H_2_O, were spiked with a cocktail of stable isotope-labelled internal standards for accurate lipid quantification and subjected to biphasic lipid extraction using chloroform-methanol. The resulting lipid extracts were analyzed by automated MS^ALL^ analysis using a robotic chip-based nanoelectrospray ion source coupled to a high-resolution Orbitrap Fusion mass spectrometer. In total, we prepared and analyzed six samples of the acid whey concentrate, three samples of the sweet whey concentrate, and three samples buttermilk whey concentrate.

The MS^ALL^ lipidomic analysis identified 5714 lipid molecules, belonging to twenty-three different lipid classes and five lipid categories (Figure 1A). Among these, 5583 lipid molecules were identified at the so-called molecular lipid species-level with annotation of individual hydrocarbon chains, and 131 lipids were identified at the so-called species-level, with assignment of the total number of C atoms, double bonds, and hydroxyl groups in all hydrocarbon chains. The majority of identifications, 4973 (89%), are molecular TAG species, which by definition are categorized as glycerolipids. The other 741 identifications are primarily glycerophospholipids (e.g., PC, PE, PS, and PI species), sphingolipids (e.g., Cer, SM, HexCer, and Hex2Cer species) as well as DAG species, which are categorized as glycerolipids. In addition, several non-esterified (free) fatty acids and sterol lipids (e.g., cholesterol, cholesteryl esters (CEs)) were also identified.

At the level of lipid quantification, expressed as pmol/mg concentrate, we monitored the aforementioned 741 lipid molecules as well as 2659 molecular TAG signatures (Appendix A). We note that individual TAG molecules were not quantified due to the technical limitation that lipidomic technology in general cannot accurately resolve and quantify complex mixtures of isomeric molecular TAG species (e.g., TAG 4:0–14:0–18:0, TAG 6:0–14:0–16:0, TAG 8:0–12:0–16:0, TAG 10:0–10:0–16:0) (Appendix A). To mitigate this, one can either quantify TAGs identified at the species-level (e.g., TAG 36:0) and disregard the molecular information about the individual FA chains that make up a particular TAG species (Appendix A). Alternatively, and as done in our dataset, individual molecular TAG signatures can be calculated that represent the abundance of every FA chain that underpin a particular TAG identified at the species-level (e.g., TAG 36:0(FA 4:0), TAG 36:0(FA 6:0), TAG 36:0(FA 8:0), TAG 36:0(FA 10:0), TAG 36:0(FA 12:0), TAG 36:0(FA 14:0), TAG 36:0(FA 16:0), TAG 36:0(FA 18:0)) (Appendix A).

To pinpoint differences between lipid abundances, we performed analysis of variance (ANOVA) with multiple hypothesis correction. For the 3400 monitored lipid molecular signatures, it was found that 2610 of these were significantly different across the concentrates by having a multiple hypothesis corrected *p*-value (i.e., q-value) < 0.01. The majority of significant differences were among molecular TAG signatures (81%), followed by PC, PE, PS, PI, DAG, Cer (all 2%), SM, HexCer, and Hex2Cer species (all 1%). Together, this result shows, as expected from the bulk lipid levels (Table 1), that the concentrates differ in terms of the overall contents of fat (i.e., TAG and DAG) and phospholipids (e.g., PC, PE, SM). Moreover, the result also highlights differences in the sphingolipids Cer, HexCer, and Hex2Cer, which are not picked up by the conventional methods.

One way to recapitulate the multitude of differences found at lipid molecular-level is to sum the abundances of all monitored lipid molecules per basis of individual lipid classes. By doing so, it is evident that the total level of TAGs shows the most pronounced difference, and that the buttermilk whey concentrate contains the highest amount of TAG lipids, followed by the sweet whey concentrate and the acid whey concentrate (Figure 1B, left panel). Similarly, the total levels of all other aforementioned lipids also show significant differences at the lipid class-level (Figure 1B, right panel).

### 3.3. Molecular Compositions of the Whey Concentrates

Although absolute lipid quantification per unit of sample material (e.g., pmol/mg concentrate) is a gold standard in analytical chemistry, it can have its disadvantages for certain types of comparative analyses. For example, it can be challenging to assess the extent to which lipids in the whey concentrates primarily stem from TAG-rich cores of MFGs or membranes of MFGMs and EVs. To address this question, the absolute lipid abundances (Appendix A) were converted to relative molar abundances by computing the proportion of every lipid molecule and molecular TAG signature relative to all monitored lipid molecules (i.e., mol% per all monitored lipids).

Evaluating relative lipid molar abundances again showed that TAG molecules were the most abundant lipids in all concentrates and represented 72% in buttermilk whey extract, 52% in sweet whey concentrate, and 37% in acid whey concentrate (Figure 1C). The second most abundant lipid category in all of the concentrates were glycerophospholipids, comprising 15%, 26%, and 33% of all lipids in buttermilk, sweet and acid whey concentrate, respectively. Within this lipid category and for all concentrates, PC species were observed to be the most abundant, followed by PE, PS, and PI species (Figure 1C). The third most abundant lipid category were sphingolipids, representing 4%, 11% and 13% of all lipids in the buttermilk, sweet and acid whey concentrates, respectively. This lipid category was primarily composed of SMs and glycosphingolipids with two hexosyl-groups (i.e., Hex2Cer) (Figure 1C). We note that lipidomic technology in general does not allow determining the stereochemical structures of the two sugar moieties, which in principle can derive from combinations of glucose, galactose, mannose, fructose and many other isomeric hexoses. Nevertheless, based on knowledge of mammalian sphingolipid biosynthesis it is highly likely that Hex2Cer in this case is synonymous with lactosylceramide (LacCer) [28]. The fourth most abundant lipid category was sterol lipids, represented primarily by cholesterol and to a lower extend CEs. Finally, it was observed that nonesterified (free) fatty acids (NEFAs) were represented as the fifth most abundant lipid category and class (Figure 1C).

Taken together, these data recapitulate the notion the buttermilk-based concentrate contains a higher proportion of MFG-related colloidal particles, and thus TAG molecules, as compared to the sweet and acid whey concentrate. Conversely, the data show that the acid and sweet whey-based concentrates have higher levels of membrane-derived colloidal particles, and thus more glycerophospholipids, sphingolipids and cholesterol as compared to the buttermilk whey concentrate. One interesting finding is that the acid whey concentrate contains a much higher level of CEs, which are apolar lipids that typically constitute the inner core of cytosolic lipid droplets (the origin of the TAG-rich core of MFGs) or secreted high- and low-density lipoprotein particles [29]. We speculate that this unique feature of the acid whey concentrate originates from lipoprotein particles. This, however, needs to be verified by further biochemical investigation.

### 3.4. Membrane Lipid Composition of Whey Concentrates

To more closely examine the molecular composition of membrane-derived material in the concentrates, we computed the molar abundance of every membrane lipid relative to all monitored membrane lipids (i.e., mol% per all membrane lipids). By doing so, we observed that membrane lipids in the concentrates were generally composed of 65% glycerophospholipids, 23% sphingolipids and 12% cholesterol (Figure 2). Interestingly, at this deeper lipid class-level, we now observed several notable differences among the concentrates.

For example, the membrane lipid composition of the buttermilk-derived concentrate featured a higher level of PE, PS, PI and Cer species as compared to that of the other concentrates. These lipids are considered to be more prominent in the ER as they are de novo synthetized here. Moreover, during biogenesis of MFGs the surrounding membrane monolayer is derived from the cytoplasmic leaflet of the ER. In contrast, the data also show that lipids considered to be more prominent in the plasma membrane, including cholesterol, SM, and Hex2Cer species, are less abundant in the buttermilk-based concentrate. Together with the biochemical and cell biological knowledge of MFG biogenesis, these data corroborate the notion that the buttermilk-derived concentrate comprises a higher level of TAG-rich cores decorated by a membrane monolayer rich in ER-derived lipids, and conversely, a lower proportion of plasma membrane-derived bilayers as compared to the two other concentrates.

The comparative analysis also shows that the membrane lipid compositions of the acid and the sweet whey-based concentrates are very similar. The most pronounced differences were a higher level of cholesterol in the acid whey concentrate, which is offset by higher levels of Hex2Cer and PC species in the sweet whey concentrate. This observation is somewhat perplexing given that these lipids can all be considered biomarkers of plasma membrane, as they are prominent constituents here. Thus, at the bulk membrane lipid class-level, it is difficult to ascertain whether the two concentrates differ in their relative proportions of MFGM- and EV-derived particles. However, based on the total levels of all lipids, including TAGs, it is evident that the acid whey concentrate has ~31% more membrane-related lipids as compared to the sweet whey concentrate.

### 3.5. Molecular Profiling of TAG Species

To explore differences at the deeper lipid molecular-level, we next turned to examine the overall profile of FA chains of all TAG molecules. To do so, we computed the relative molar abundances of individual FA chains attached to all detected TAG molecules and normalized these values to the total molar abundance of all monitored FA chains. By doing so, we effectively mimic the output of conventional methods based on preparative fractionation of TAGs followed by gas-chromatography-based fatty acid profiling but do so much more time-effectively and less labor-intensively.

This in silico analysis demonstrated that TAG species in the three concentrates primarily featured saturated and monounsaturated long-chain FAs (e.g., 16:0, 18:1, 14:0, 18:0) as well as saturated short- and medium-chain FAs (e.g., 4:0, 6:0, 8:0, 10:0, 12:0). In addition, many other FA chains where observed, including 2:0 and odd-numbered long-, medium-, and short-chain FAs (e.g., 19:1, 17:0, 15:0, 13:0, 11:0, 9:0, 7:0, 5:0) (Figure 3). Notably, the overall levels of very long-chain FAs, such as 22:0, 23:0 and 24:0, were very low (~1–3% of the total level of FA chains in TAGs) (Figure 3B).

Comparison of FA profiles showed that the three concentrates were highly similar (Figure 3A). This indicates that the lipid metabolic machinery responsible for TAG production in cow milk-producing cells has a rather reproducible product-specificity in terms of the TAG molecules it produces (as well as other lipid molecules, see section below). Nevertheless, some less pronounced, yet systematic differences were observed. Especially, the buttermilk whey concentrate showed higher levels of 18:1 and 4:0 chains as well as lower levels of 16:0 and 14:0 chains as compared to the two other concentrates (Figure 3A). We speculate that these subtle differences relate to variability in the input raw cow milk and hereunder the time-of-year when the concentrates were produced.

### 3.6. Similar Profiles of Sphingolipids in the Whey Concentrates

While exploring molecular differences, we were surprised to discover that the molecular profiles of sphingolipids were highly comparable. As expected, and based on knowledge of sphingolipid metabolism in mammals, we found that the majority of sphingolipids in the three concentrates contained an 18:1;2 sphingoid base (i.e., C_18_-sphingosine), ranging from ~47% of all ceramide species to 69% of all HexCer species (Figure 4A). Conversely, and surprisingly, the proportions of more atypical sphingoid base moieties were relatively high, where 13% to 22% of all sphingolipids comprise a 16:1;2 moiety, ~5% have an odd-numbered 17:1;2 moiety, and ~3% have a 19:1;2 moiety (Figure 4A). These data indicate that the lipid enzymatic machinery responsible for sphingolipid production in cow milk-producing cells not only uses the serine and 16:0-CoA for synthesis of canonical C_18_ sphingoid bases but also uses 14:0-CoA, 15:0-CoA, and 17:0-CoA to produce the atypical C_16_, C_17,_ and C_19_ sphingoid bases, respectively.

The molecular profile of FA chains attached to the different sphingolipid classes also showed a highly reproducible pattern. In general, the data show that saturated FA chains where the most prominent, including 16:0, 22:0, 23:0, and 24:0 (Figure 4B,C). Notably, a characteristic signature of sphingolipids in cow milk is the relatively high proportion of the odd-numbered 23:0 chain, which is typically less abundant than well-known and more commonly detected 22:0 and 24:0 chains. Another interesting finding is that sphingolipids with an 18:1;2 sphingoid base are biosynthetically coupled to all FA chains whereas sphingolipids a 16:1;2 moiety are more frequently coupled to very long-chain 22:0, 23:0, and 24:0 residues and to lesser extend to the long-chain 16:0 residue (Figure 4B,C). This indicates that the underlying enzymatic machinery responsible for sphingolipid synthesis selectively couples very long-chain FA to the shorter 16:1;2 sphingoid base and, conversely, is more promiscuous with regard to the chain-length of FAs that are coupled to the canonical 18;1;2 sphingoid base.

## 4. Conclusions

Here, we comprehensively characterized the molecular lipid composition of three whey concentrates, obtained using different production pipelines. Overall, our lipidomic data shows that the three whey concentrates have unique lipid compositions that distinguish these at the bulk lipid-level as well as on the level of individual lipid molecules. Besides uncovering a multitude of lipid signatures that distinguish the concentrates, our data also reveal that the molecular profile of especially sphingolipids is astonishingly constant across the three whey concentrates. Altogether, our study provides a valuable quantitative resource of individual lipid molecules in whey concentrates, which can be used as reference in the field and as a supplementary ingredient list for documenting the nutritional value of functional foods containing dairy-based concentrates.

In addition to reporting a detailed lipidomic resource with unprecedented coverage for whey concentrates, we also show that the resource prompts new insights into the lipid biosynthetic machinery of milk-producing cells and the overall proportions of MFG-related particles as well as MFGM- and EV-derived membrane particles. In the perspective of lipid production by milk-producing cells, it is rather remarkable that the molecular compositions of sphingolipids are close to identical across all the concentrates albeit these have been produced by different cows at different time-points of the year. Furthermore, the high abundance of sphingolipids with 23:0 chains seem to be a distinct signature of milk-based products since 23:0-containing sphingolipids are typically not detected at high levels in biopsies of humans and other animals. Notably, other monitored lipids, including TAG species, in the whey concentrates do not contain significant amount of 23:0 chains. Hence, monitoring of the 23:0-containing lipids in biopsies from human and other animals might be used as a proxy for the uptake and accretion of milk sphingolipids.

Our data, and the underlying MS^ALL^ technology, are in principle able to accurately measure the ratio between the lipid amounts derived from MFGs as well as the sum of membrane lipids from MFGMs and EVs. This can be done using the molar abundances of TAGs and other neutral lipids as proxies for the amount of MFGs, and the levels of de facto membrane lipids, including PC, PE, PS, PI, SM, Hex2Cer, and cholesterol, as proxies for the content of MFGM- and EV-derived lipids. Whether the MS^ALL^ technology, or other lipidomic methods [30], can be used to further disentangle the proportions of membrane lipids from EVs and MFGMs in whey concentrates is at the present unclear. Encouragingly, the data revealed significant, yet rather subtle, differences in the ratios of various membrane lipids. For example, the acid whey concentrate, with the lowest content of TAG-rich cores, featured a higher level of cholesterol as compared to the sweet whey concentrate, which in turn had a higher proportion of Hex2Cer (i.e., LacCer) species. It is tempting to speculate that these systematic differences could be due to the different mechanisms of secretion of EVs and MFGs. Here, the mechanism of EV secretion could be more dependent on the level of cholesterol at the plasma membrane whereas the secretion and shredding of MFGs is more dependent on LacCer. Exploring this notion obviously warrants more research. However, it is known that enveloped viruses are secreted from epithelial cells with a relatively high level of cholesterol in their surrounding membrane [31].

In summary, our study provides a high-quality resource of lipid molecular abundances in different whey concentrates and lays the groundwork for using in-depth MS^ALL^ lipidomic technology for profiling the nutritional value of milk products and functional foods containing dairy-based concentrates.

## Figures and Tables

**Figure 1 biomolecules-14-00055-f001:**
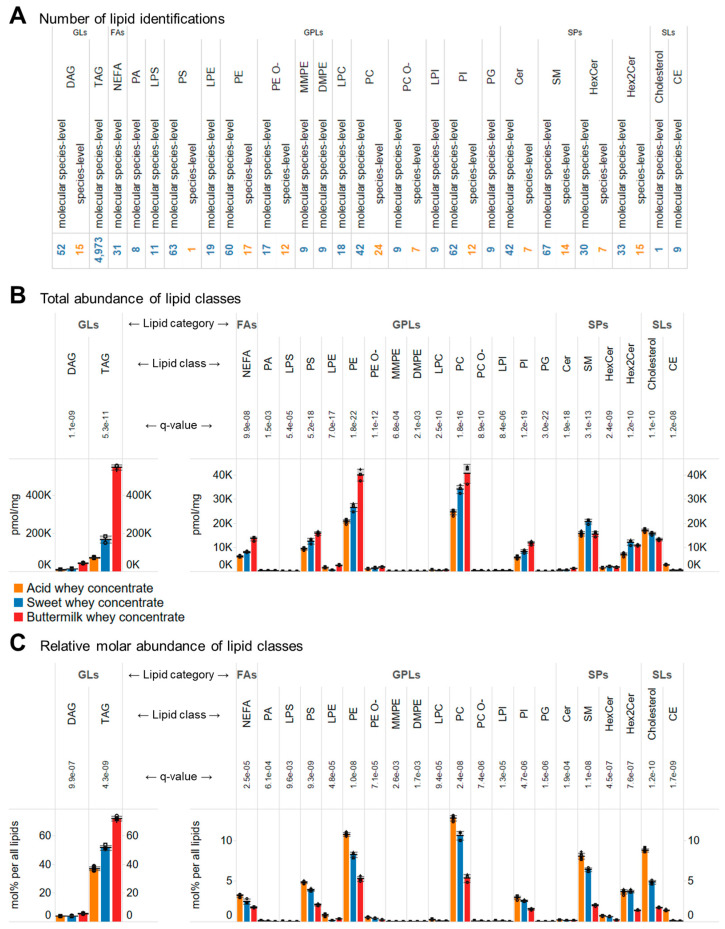
Lipidome coverage and overall lipid class levels. (**A**) Number of identified lipid molecules. (**B**) Absolute abundances of lipid classes expressed as pmol/mg concentrate. (**C**) Relative molar abundances of lipid classes expressed as mol% per all lipids. Data represent mean ± SD (*n* = 6 for the acid whey concentrate and *n* = 3 for the other whey concentrates).

**Figure 2 biomolecules-14-00055-f002:**
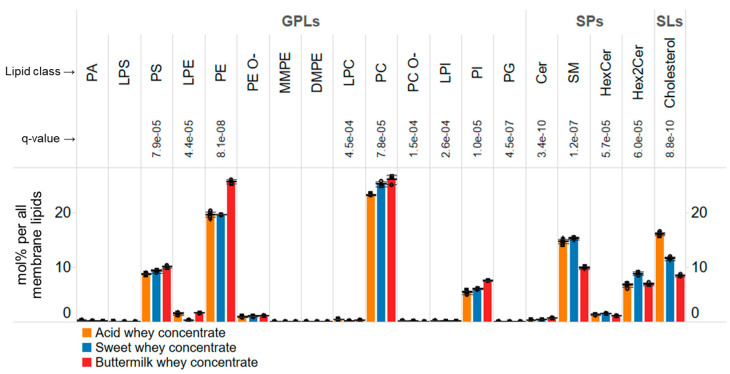
Overall profile of membrane lipids. Molar abundances of membrane lipid classes expressed as mol% per all membrane lipids. Data represent mean ± SD (*n* = 6 for the acid whey concentrate and *n* = 3 for the other whey concentrates).

**Figure 3 biomolecules-14-00055-f003:**
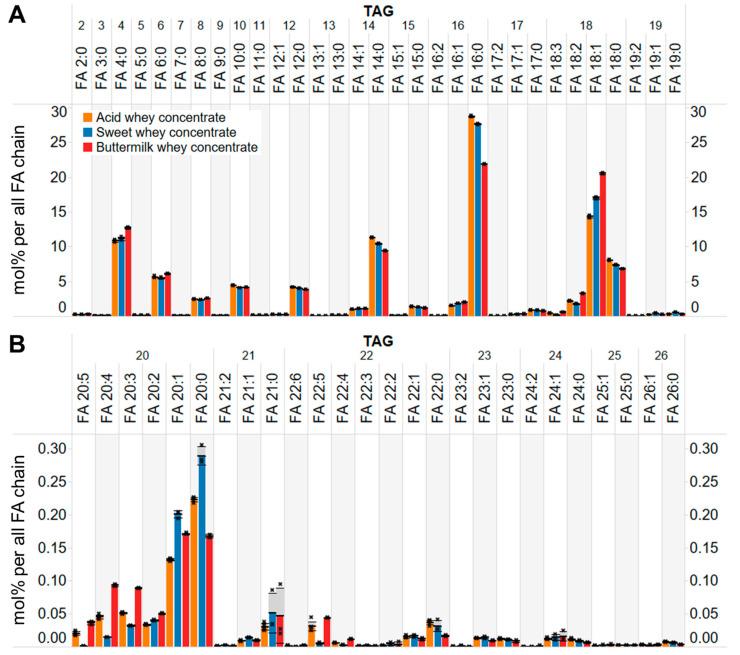
Total FA profile of TAG molecules. (**A**) Molar abundance of short, medium and long-chain FAs attached to TAG molecules of the three concentrates. (**B**) Molar abundance of very long-chain FAs attached to TAG molecules of the three concentrates. Data represent mean ± SD (*n* = 6 for the acid whey concentrate and *n* = 3 for the other whey concentrates).

**Figure 4 biomolecules-14-00055-f004:**
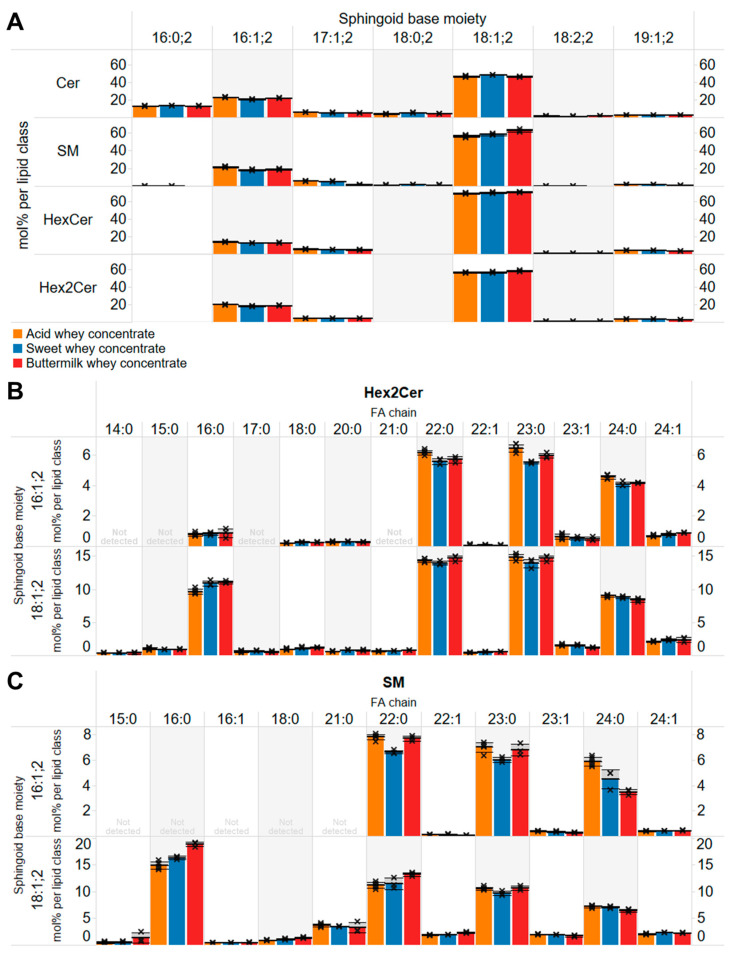
Molecular profile of sphingolipids in whey concentrates. (**A**) Overall profile of sphingoid base moieties. (**B**) Profile of molecular Hex2Cer species. (**C**) Profile of molecular SM species. Data represent mean ± SD (*n* = 6 for the acid whey concentrate and *n* = 3 for the other whey concentrates).

**Table 1 biomolecules-14-00055-t001:** Overall composition of whey concentrates.

	% of Concentrate ^1,2^					% of Total Lipid	
	Dry Matter	Protein	Total Phospholipid ^3^	Total Lipid ^4^	Total Fat ^5^	Total Phospholipid	Total Fat
Acid whey concentrate	96	76	5	11	6	46	54
Sweet whey concentrate	96	70	8	22	14	35	65
Buttermilk whey concentrate	97	34	9	34	25	26	74

^1^ Percent values are based on weight (g) per total weight (g). ^2^ Percent values are based on bulk gravimetric analyses and ^31^P-NMR spectroscopy. ^3^ Total phospholipid is a proxy for the sum of all glycerophospholipids (e.g., PC, PE, PS, PI) and SM. ^4^ Total lipid is a proxy for the sum of all lipids. ^5^ Total fat is used as a proxy for the total amount of TAG. It is estimated as total lipid minus total phospholipid.

## Data Availability

Raw lipidomic data are available upon reasonable request.

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
