# Peer review of "Lipidomic Characterization of Whey Concentrates Rich in Milk Fat Globule Membranes and Extracellular Vesicles"

_biomolecules, 2023, doi:10.3390/biom14010055_

Round 1

Reviewer 1 Report

Comments and Suggestions for Authors

Biomolecules-2768545

Lipidomic characterization of whey concentrates rich in milk fat globule membranes and extracellular vesicles Special Issue: Fatty Acids in Natural Ecosystems and Human Nutrition

This manuscript characterises the lipids in three milk-derived whey concentrates. The analytical results in this paper are of the highest standard, and using higher resolution instrumentation that is usually used. . What is impressive is that the species have not only been identified but also well quantitated.

Thus, my main suggestion is that the context of the system from which the concentrates are derived should be better explained to enable the data to be more easily understood. These concentrates are derived from dairy streams, and so rather than being unique organelles they have the mixed properties of the diverse composition of the streams from which they are derived. For instance, the buttermilk is derived from a cream stream which is high in fat so the higher lipid content is to be expected in the aqueous stream. In that sense it may have been better to concentrate on the variations within the lipid classes as well as between the lipid classes. This would focus on difference which may have metabolic sources.

Thus an explanation of the processing involved in the product of the various streams showed be included and used as a basis of the discussion on the variation

The difference between MFGM and EV is not well explained in terms of the results in this paper. Lines 49-63 have a general explanation but much of the paper talks about MFGM + EV. Line 424 and on, describe the difference as based on the amount of TAG in the stream but to what extent is this just a consequence of the dairy separation systems used. The authors imply that EV do not contain TAG. If so, this could be more clearly stated.

Buttermilk whey concentrate, just for my understanding is this the buttermilk produced from making whey butter from whey cream with the whey buttermilk side stream or is it from whole milk cream into butter.

Line 325 worth explaining that the MFGM contains a trilayer, made up of the two parts described.

Worth stating what mode (neg or positive) was used for the difference lipid classes and in Figure 4, the conditions which allowed the identification of the sphingoloid base.

May be beneficial to add the following reference
Brink, L. R. et al., (2020). Omics analysis reveals variations among commercial sources of bovine milk fat globule membrane. J Dairy Sci, 103, 3002-3016.

Reviewer 2 Report

Comments and Suggestions for Authors

Journal: MDPI_biomolecules 

Title: Lipidomic Characterization of Whey Concentrates Rich in Milk Fat Globule Membranes and Extracellular Vesicles 

Manuscript ID: biomolecules_2768545 

Comments 

This study investigated the “Lipidomic Characterization of Whey Concentrates Rich in Milk Fat Globule Membranes and Extracellular Vesicles”. Whey concentrates, enriched with these beneficial lipids, find applications in infant formulas to emulate human milk and in medical nutrition products aimed at enhancing the metabolic fitness of adults and the elderly. This study presents a comprehensive and quantitative lipidomic analysis of various whey concentrates. This study provides a comprehensive understanding of lipid molecules within whey concentrates, offering valuable insights for the development of medical nutrition products. However, to submit this paper to the MDPI Biomolecules Journal, the following comments need to be revised.

1.      The nutritional characteristics of whey concentrates (sweet, acid, buttermilk) were reported based on the analysis of lipid molecules. However, there is no mention of the sampling method or the number of samples (n=?) representing the results for each type of whey concentrate. Please verify this information.

2.      The lipid composition of whey concentrate samples can vary depending on the processing methods associated with the type of raw material. It raises concerns whether the three samples adequately represent the lipid content for meaningful comparisons.

3.      The representation of the analysis results in all figures is not clearly defined. Please present them in tables, indicating mean ± SD (n=?) and highlighting significance where applicable.

4.      Even when considering lipid molecules, the extraction and analysis methods vary for each component. However, in this paper, there is insufficient description of the analysis methods, despite the importance of quantifying each lipid component. For example, details on p-NMR analysis, lipid extraction methods for each component, and validation for quantification through these methods need to be addressed.

Reviewer 3 Report

Comments and Suggestions for Authors

Dear Editor and Authors,

I send you my review about the article “Lipidomic characterization of whey concentrates rich in milk fat globule membranes and extracellular vesicles”.

The scope of the paper, as reported in the aim was to develop a comprehensive, quantitative study of individual lipid molecules in cow milk-derived whey concentrates. In my opinion, the paper is well structured and it is, also, interesting. However, it need of some little improvements that I report below.

The introduction is well written, but it should be better explained the originality of this article. In my opinion, in this chapter, the comparison with the other articles, present in the references, it should be deepened. Moreover, should be better explained the difference among this research and the oldest one.

Materials and methods is well written and well structured.

Furthermore, the methods used for statistical analysis of the data is well showed.

Nevertheless, in the Paragraph 2.2, “Procurement of whey concentrates”, should be better described the procedures used to produce the milk whey and the whey concentrated.

For example, how was produced the milk whey (quantity of milk used, pH of precipitation of casein, how is separated whey form precipitate) and how and how many was concentrated the whey.

The results is very well presented and they are very well discussed, also in comparison to the data reported in the literature.

Moreover, data were well show in figures that they are adequate to the results.

Moreover, the conclusions of the paper result adequate to the results showed and they satisfy the aim of the Article.

However the conclusion should not repeat or summarise the discussion of data but they should be focused on the answer of the aim.

Nevertheless, conclusion should include some personal comments of the Authors.

In this regard, I would suggest that the authors report their opinion on the impact that the results of their methodology could have.

Best regards

Round 2

Reviewer 2 Report

Comments and Suggestions for Authors

Journal: MDPI_biomolecules 

Title: Lipidomic Characterization of Whey Concentrates Rich in Milk Fat Globule Membranes and Extracellular Vesicles 

Manuscript ID: biomolecules_2768545_R1 

Comments

This manuscript was properly edited by the authors. The edited manuscript is suitable for publication in “biomolecules” Journal.
